# Visual Perception of Regularity and the Composition Pattern Type of the Facade

Michał Malewczyk *, Antoni Taraszkiewicz and Piotr Czyż

Faculty of Architecture, Gdansk University of Technology, Gabriela Narutowicza 11/12, 80-233 Gdańsk, Poland
* Correspondence: michal.malewczyk@pg.edu.pl

**Abstract:** The present study investigates the degree of visual regularity perceived by viewers in architectural compositions, specifically concerning the type of pattern used. The research is grounded in psychological and neuropsychological universal determinants of visual perception, particularly the perception of visual regularity. The study is based on an empirical survey that involved 48 participants who rated various compositions on a Likert scale. The stimuli presented consisted of a typology of compositional patterns of facades of Polish multifamily buildings developed by Malewczyk, Taraszkiewicz, and Czyż in 2022. The survey results were subjected to statistical analyses, which revealed a clear relationship between the type of composition and its perceived regularity. This implies that architects can predict the perceived regularity of a composition based on its type, which is crucial since visual regularity is closely linked to the sense of spatial order and aesthetic value. Both of these aspects are vital for perceiving architecture as a built environment. The study highlights the significance of visual perception in architectural design, particularly how the public perceives composition types.

**Keywords:** architecture; aesthetics; cognitive evaluation; environmental psychology; visual analysis





## 1. Introduction

Christopher Alexander's Pattern Language theory is one of the most influential theories in architecture. Initially, it consisted of 253 patterns, which provided solutions to design problems in urban planning, building design, and construction. Malewczyk, Taraszkiewicz, and Czyż [1] identified another pattern related to the composition of elements on the facades of multifamily buildings. This pattern includes six different compositions (shown in Figure 1) that describe all possible arrangements of components, such as windows or balconies on the facade of a multifamily building. However, it is essential to understand how the public perceives these compositions. This study focuses on the visual perception of these composition types. By parameterizing the composition types in terms of their perception by potential viewers, this pattern can be applied with the knowledge of how a statistical viewer will perceive an architectural object based on a particular composition type.

Visual beauty is a crucial factor in architectural design. But how does a building's composition relate to its aesthetic appeal? The following subsection, "Beauty and Composition", attempts to answer this question.

### 1.1. Beauty and Composition

Several seminal works provide essential context and analysis to deepen the understanding of how the arrangement and composition of architectural elements influence aesthetic appreciation, drawing on psychology and the mathematical principles of fractals.

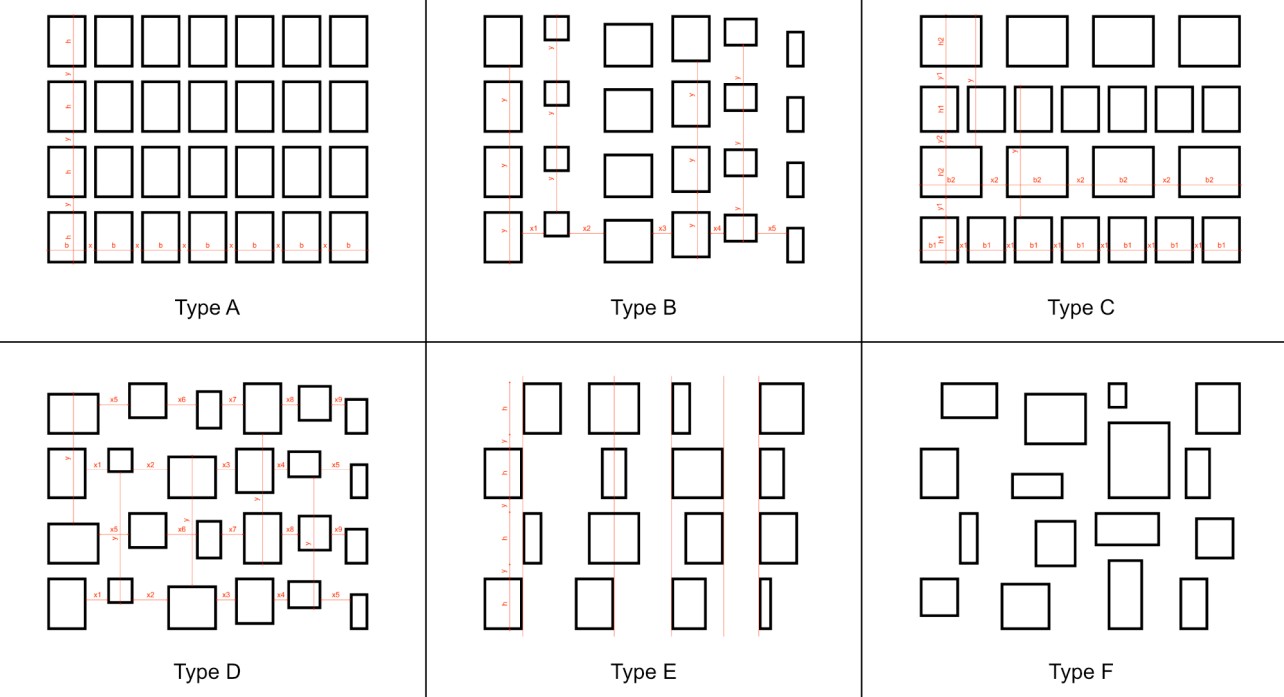

**Figure 1.** Schemes representing compositional pattern types [1].

Lang's study of the behavioral sciences in environmental design [2] provides valuable insights into the psychological foundations of spatial perception, which can enhance discussions on how architectural composition impacts aesthetics. Arnheim's research on the dynamics of form [3] offers a comprehensive theoretical framework for understanding how architectural shapes and structures affect human aesthetic perception. Bovill [4] introduces fractal geometry as a tool to appreciate the complexity and beauty of architectural designs, providing a scientific approach to understanding the patterns that underlie aesthetic preferences.

Mandelbrot's seminal work on fractals [5] provides the mathematical foundation for discussions on complexity and regularity in architectural patterns, which are essential for understanding the nuances of architectural aesthetics. Ramachandran and Hirstein [6] propose a neurological basis for aesthetic experiences, bridging the gap between the perception of architectural forms and the underlying brain processes. Joye's [7] discourse on biophilic design adds another dimension to architectural aesthetics by highlighting the innate human affinity for nature and natural patterns within built environments.

Zeki [8] explores the neural mechanisms involved in art and aesthetic perception, providing a fundamental understanding of how the brain processes the forms and compositions of architecture. Finally, Ching [9] introduces the essential architectural design elements, including form, space, and order. Knowledge of these elements is crucial in discussions about how architectural composition contributes to buildings' overall aesthetic and visual coherence.

Together, the references provide a comprehensive understanding of the relationship between architectural elements' design and their aesthetic appeal. They highlight the importance of studying the correlation between specific design types and architectural objects' visual and aesthetic characteristics.

Those objects, especially multifamily buildings, are composed of many standardized elements, such as walls or windows. Each of these elements forms a particular whole (the building), which, as a collection, interacts with the viewer and evokes specific aesthetic sensations. According to Birkhoff and Eysenck's theory, the aesthetics of such objects are measurable and depend on the parameter C, which is the complexity of the object, and the

parameter O, which is the regularity of the whole system and the individual elements. The relevance and scientific value of these methods have been repeatedly confirmed [10,11], although their application in the context of architecture has so far been mainly limited to existing objects [12]. However, the relationship between aesthetics and regularity was already pointed out by the predecessor of empirical aesthetics—Gustav Theodor Fechner [13]. The results of research in the field of experimental psychology also indicate the relationship between visual regularity and the aesthetics of an object. Four experiments by the team of Pecchinenda, Bertamini, Makin, and Ruta [14] should be mentioned. The results clearly show a preference for patterns and symbols characterized by bilateral symmetry over objects without this feature. Studies on preferences for geometric patterns also show a similar tendency. Patterns based on fractals, i.e., highly ordered (regular) structures, are considered more aesthetic than random patterns [15–17].

Furthermore, abstract patterns are more positively associated when their regularity is greater [18]. In the context of architecture, the same validity is confirmed by the study of the team of Malewczyk, Taraszkiewicz, and Czyż [19]. Statistical analyses of the survey results clearly show that more regular facades are perceived as more aesthetic. Other studies addressing these issues are also worth noting. Sussman and Ward [20], on the basis of studies using an eye-tracking apparatus, indicate that people tend to overlook empty facade spaces and uninteresting glass facades. Extreme visual regularity can be combined with monotony, and this monotony can negatively affect the audience of such architecture [21]. Very high visual repetition can even cause the atrophy of gray brain cells [22,23]. There are also positions that suggest the need to find a balance between overwhelming regularity and exaggerated, chaotic irregularity [24–26].

In the case of an architectural object, the composition is responsible for visual regularity and, therefore, the parameter O—visual order (in Birkhoff and Eysenck's formula). According to Rob Krier [27], the facade is the most important architectural element determining its aesthetic value. Secondly, for both Arnheim [3] and Alexander [12], the order is created by the geometry of the facade and, thus, among other things, by its composition. Thirdly, like regular structures, ordered structures display a strong sense of wholeness and evoke a robust perceptual response [28–30]. Attention should also be paid to the proposal of the team of Meddahi and Boussur [31] regarding the parametric description of the facade order in the context of the Eysenck method. This proposal makes the order of the facade dependent on features such as symmetry, repeatability, or coherence, among others. According to the authors of this study, these features can describe any regular composition, which also suggests a relationship between the O (order) parameter and the facade's composition.

In conclusion, the cited theories and empirical results demonstrate the relationship between visual regularity and aesthetic preferences. It should be noted, however, that these conclusions are based on studies based on the traditional understanding of beauty, for Western cultures, as pleasure derived from aesthetic sensations. Such an understanding of beauty corresponds, however, with the critical element for this study, which is the typology of composition developed by Malewczyk, Taraszkiewicz, and Czyż [1], referring to the facades of Polish multifamily residential buildings. Culturally, Poland is a Western country, and analyses of this typology in the context of aesthetic categories should be based on the Western way of understanding these categories. The general correlation from the quoted sources is that the more regular a visual stimulus is, the more aesthetically pleasing it is perceived to be.

In the case of buildings, visual regularity is determined by the composition of the elements. Therefore, it is necessary to determine the correlation between composition type and the level of visual regularity, which makes it possible to assume how the six defined composition types [1] correlate with the aesthetic qualities of an architectural object. Visual regularity is a crucial concept that will be analyzed in the following subsection.

## 1.2. Regularity

Regularity is a multidimensional concept. However, the authors suggest that, for the purpose of this study, it should be considered as a continuum, with completely regular (ordered) systems at one end and completely random (chaotic) systems at the other. This way of thinking about visual regularity is more than just a theoretical assumption. The research by Kubilius, Wagemans, and Op de Beeck [32] confirms a linear change in the response of specific brain areas with a change in stimulus regularity. The way the brain responds is also essential for these considerations. Previous studies have shown the appearance of responses to regular stimuli and no reactions to irregular stimuli [28–30]. Irregularity is, therefore, not something separate that would trigger an opposite reaction in the brain. Hence, the regularity of the stimulus should be considered as the degree of intensity of this feature, where complete irregularity means a lack of regularity. However, Rudolf Arnheim reached the same conclusions intuitively [3], which only proves the possibility of behavioral analysis of these phenomena.

The studies conducted so far also demonstrate the relationship between regularity (or the lack thereof) and the size of the pattern elements, the distances between the elements, and their position [33]. Also worth noting are the studies on the perception of symmetry, which is an attribute of the most regular compositions. For example, they demonstrate the automaticity of the symmetry perception process, for which specific brain structures are responsible [34]. Therefore, according to the authors, it can be assumed that the perception of regularity is an objective phenomenon resulting from the solid biological basis of this process. It is also likely that the level of regularity of certain stimuli will have a similar effect on the perception of different recipients.

Nevertheless, the method's characteristics should also be considered when assessing the degree of regularity. This method is not sensitive to composition distortion at 5% of the distance between the elements. Therefore, slight shifts in the elements do not affect the system's perception. In addition, the number of components of the set analyzed is significantly limited and may depend on the individual's characteristics [35].

Despite the existence of many studies on the perception of regular and irregular patterns or the perception of symmetry, this issue has yet to be investigated in the context of architecture. The studies cited in Section 1.1, proving observers' overlooking of monotonous or empty facade elements [20], the negative perception of monotonous architecture [21], and its harmful effects on the human brain [22,23], point out that this problem is universal to the human race in general, and also with regard to architecture. The question then remains, what do we perceive as regular and as irregular?

## 1.3. Aims of the Study

Modern architecture, predominantly residential, very often operates with aesthetics based on visual irregularity [36]. At the same time, in the authors' opinion, this topic has yet to be studied in sufficient depth. Designers' lack of knowledge of how their artwork will interact with potential viewers can result in artwork that is mismatched to the needs of its users. According to the sources quoted, visual regularity correlates positively with aesthetic preferences. Moreover, regularity is perceived by the human race in general in a very positive way, which is due to neuropsychological determinants. In addition, the extension of empirical research on the perception of regularity in the context of architecture and such elementary particles of it as composition provides an opportunity to explore a new area of study and, at the same time, develops the possibility of shaping the aesthetics of architectural objects "from below" (von Unten—according to the empirical tradition). This approach aligns with the contemporary interest in experimental aesthetics and neuroaesthetics in architectural design.

The purpose of this study is to determine, by empirical means, the degree of visual regularity depending on the composition type of the facade, as defined by the authors Malewczyk, Taraszkiewicz, and Czyż [1]. As mentioned earlier, determining how the statistical viewer perceives the composition would allow for projections based on these

perceptions more specific to the potential viewer. This study aims to answer the following research questions:

1. Is there a relationship between the type of facade composition and its degree of regularity?
2. Which types of composition are the most regular, and which are the least?
3. Does the degree of regularity of the facade composition depend only on the type of composition?
4. Has the presentation of stimuli in different configurations influenced the determination of the degree of regularity?

The authors focus on examining compositional patterns in themselves, isolated. This approach excludes interactions between the architecture's various components, such as the overall form of the object, scale, material, or others. The authors also decided to eliminate a factor related to how architecture is perceived, which is subjected to perspective distortions in natural perception. In the authors' opinion, learning about such raw results and confronting them in the future with the results of a broader study that considers all those aspects of architecture and its perception that were excluded in the present study will become of additional value. It will allow us to explore the principles of interaction between the components of architectural objects, including architectural composition, which is the subject of the analyses contained in this study.

This study is based on the statistical analysis of a survey conducted on a random group of 48 respondents. The Materials and Methods, Results, and Discussion chapters describe the study, its results, and its conclusions.

## 2. Materials and Methods

### 2.1. Participants

Since the study was conducted based on universal neuropsychological determinants of the perception of visual regularity, which are expected of the entire human race (see Section 1.2), the minimum sample size (N) was set at 30. The above assumption was made based on the Central Limit Theorem. This is the minimum number, justifying a normal distribution in the statistical analyses of the collected survey results. According to the literature, the minimum sample size for survey scales (for example, the Likert scale used in this article) is $31.61 \pm 2.33$ ($p < 0.05$) [37]. The sample size in this described study meets both requirements and is representative of the general Polish population.

Respondents were randomly recruited from first-year Architecture and Spatial Management students studying at the Faculty of Architecture at the Gdańsk University of Technology. Participation in the survey was anonymous and voluntary.

### 2.2. Materials

#### 2.2.1. Stimuli

For the survey, 12 abstract graphics were prepared, representing the compositions of black rectangles on a white background. All graphics are based on a square of the same size (1200 × 1200 px). The compositions were designed based on the definition of types of facade compositions for contemporary Polish residential and multifamily buildings [1]. The specified six types (marked with letters from A to F) cover the entire spectrum of possible types of arrangements of architectural elements (such as windows or balconies) on the facades of multifamily buildings. Based on the definitions of these six composition types, it is possible to create an infinite number of composition variations, even selecting different sizes of elements. However, the authors of this study decided to prepare two stimuli for each of the six composition types. A total of 12 stimuli were ready. In limiting the number of stimuli to 12, the authors were guided by a desire to minimize the risk of falsifying results due to respondents' lassitude.

**Stimuli A1 and A2.** These compositions (Figure 2) were designed according to the definition of composition type A [1].

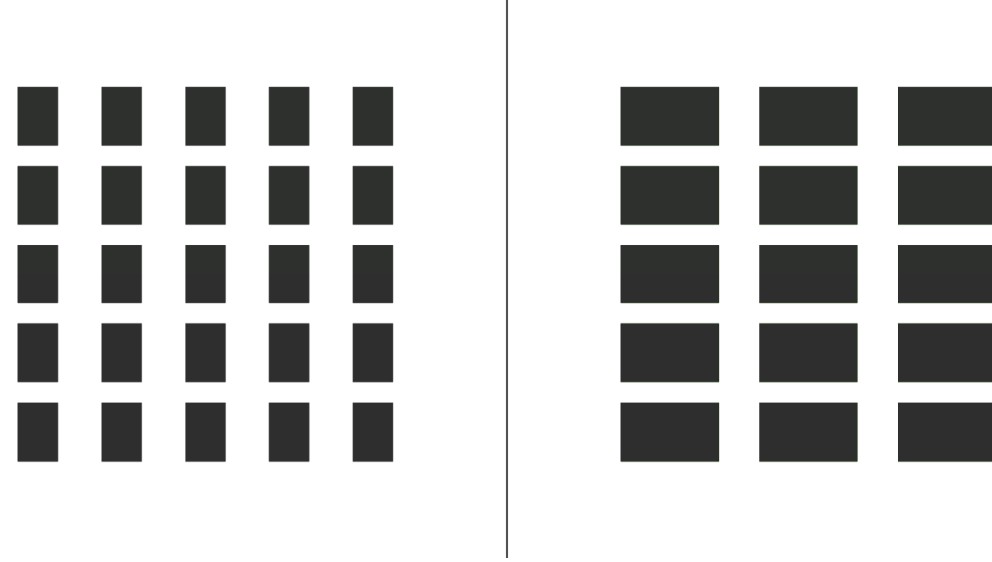

**Figure 2.** Compositions A1 (**left**) and A2 (**right**) presented during the research.

Compositions A1 and A2 differ in the size and quantity of the composition elements.

**Stimuli B1 and B2.** These compositions (Figure 3) were designed according to the definition of composition type B [1].

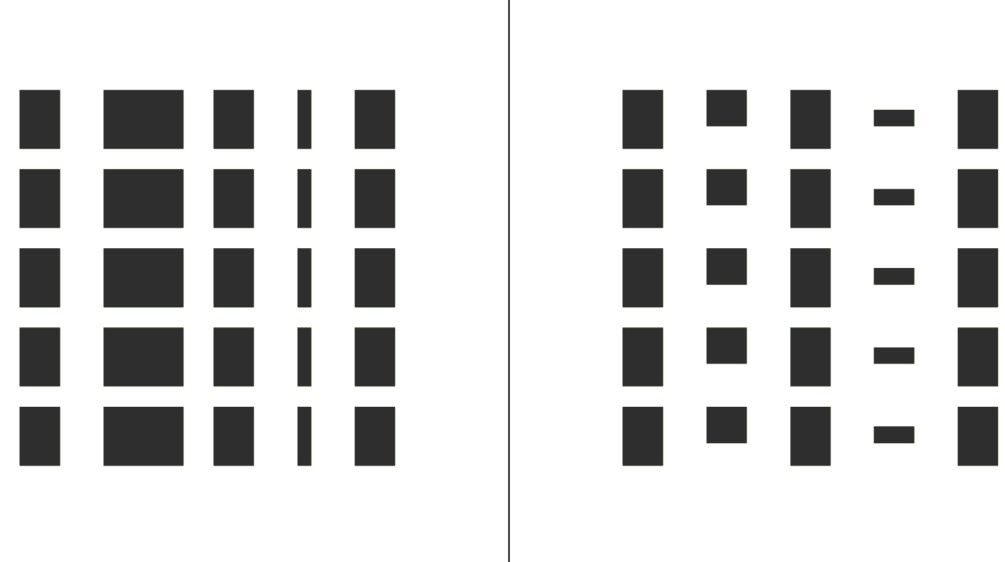

**Figure 3.** Compositions B1 (**left**) and B2 (**right**) presented during the research.

Composition B1 has elements with different widths but constant heights. Composition B2 has elements of the same width but different heights. The horizontal distances between the composition elements are different in the case of Composition B1 (two types) and the same in the case of Composition B2. The vertical spacing between the composition elements is the same for both compositions.

**Stimuli C1 and C2.** These compositions (Figure 4) were designed according to the definition of composition type C [1].

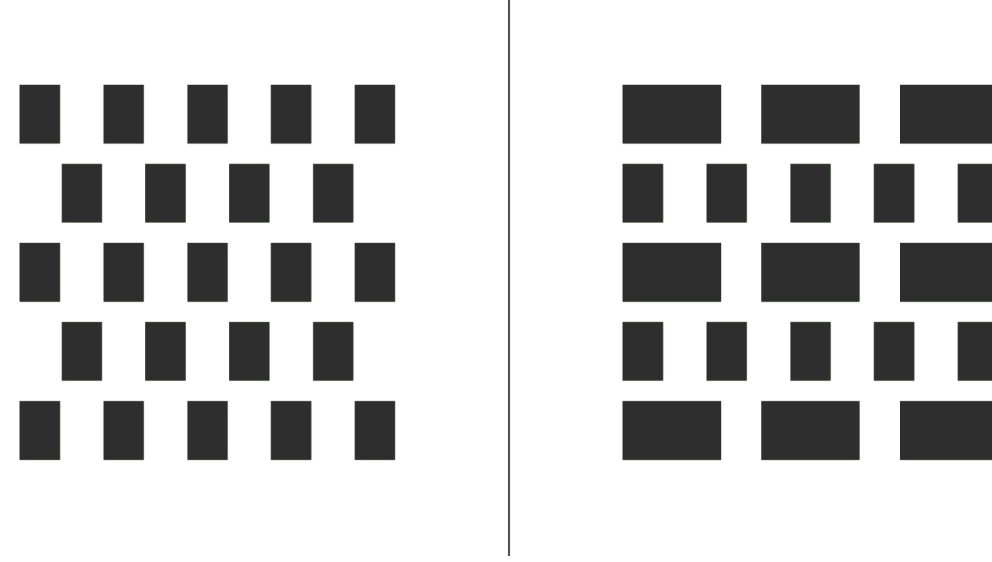

**Figure 4.** Compositions C1 (**left**) and C2 (**right**) presented during the research.

Composition C1 is based on one type of compositional element, and Composition C2 is based on two types of compositional elements that differ in width. The horizontal and vertical distances between the compositional elements are the same for both compositions.

**Stimuli D1 and D2.** These compositions (Figure 5) were designed according to the definition of composition type D [1].

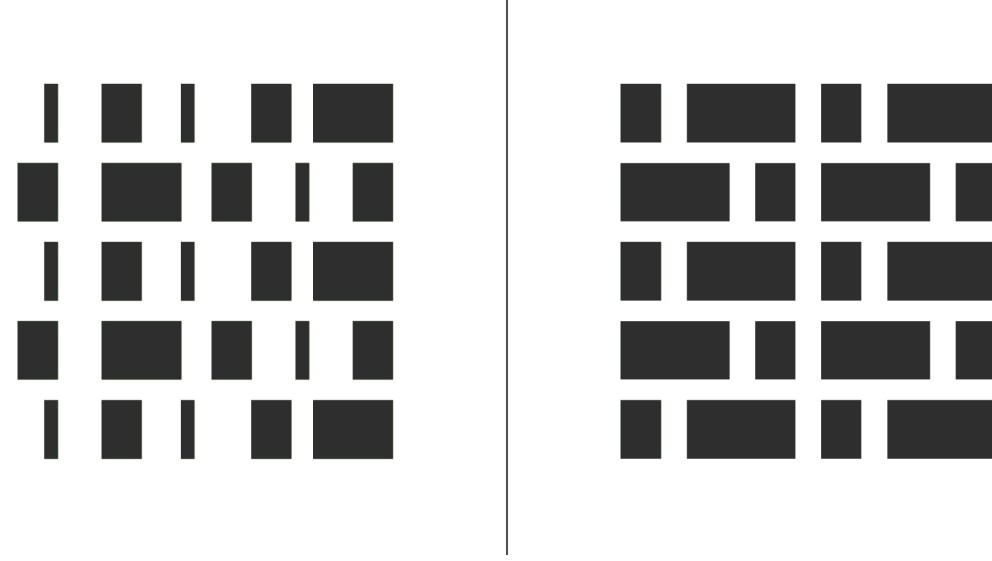

**Figure 5.** Compositions D1 (**left**) and D2 (**right**) presented during the research.

Composition D1 was based on three types of compositional elements with different widths but equal heights. In the odd lines, there are three different horizontal spacings between compositional elements, and in even lines, there are two types of distances. Composition D2 was based on two types of composition elements with different widths but the same heights. The horizontal spacing between compositional elements is the same in odd and even lines. The vertical intervals between the composition elements are the same in both compositions.

**Stimuli E1 and E2.** These compositions (Figure 6) were designed according to the definition of composition type E [1].

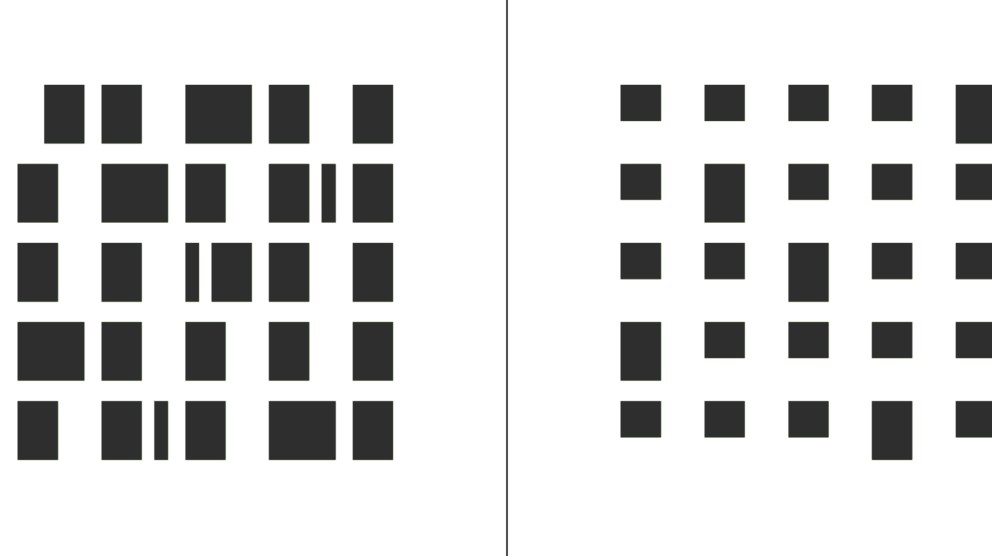

**Figure 6.** Compositions E1 and E2 presented during the research.

Composition E1 is based on three types of composition elements with different widths but the same heights. The horizontal distances between the compositional elements are varied and distinct in the following rows of compositional elements, and the vertical distances between the compositional elements are the same. Composition E2 is based on two types of compositional elements with equal widths but different heights. The horizontal distances between the compositional elements are the same, but the vertical distances between the elements are varied and different in successive columns of compositional elements. In both compositions, the compositional elements are arranged so that the side edges of these elements form lines (vertical in the E1 composition and horizontal in the E2 composition).

**Stimuli F1 and F2.** These compositions (Figure 7) were designed according to the definition of composition type F [1].

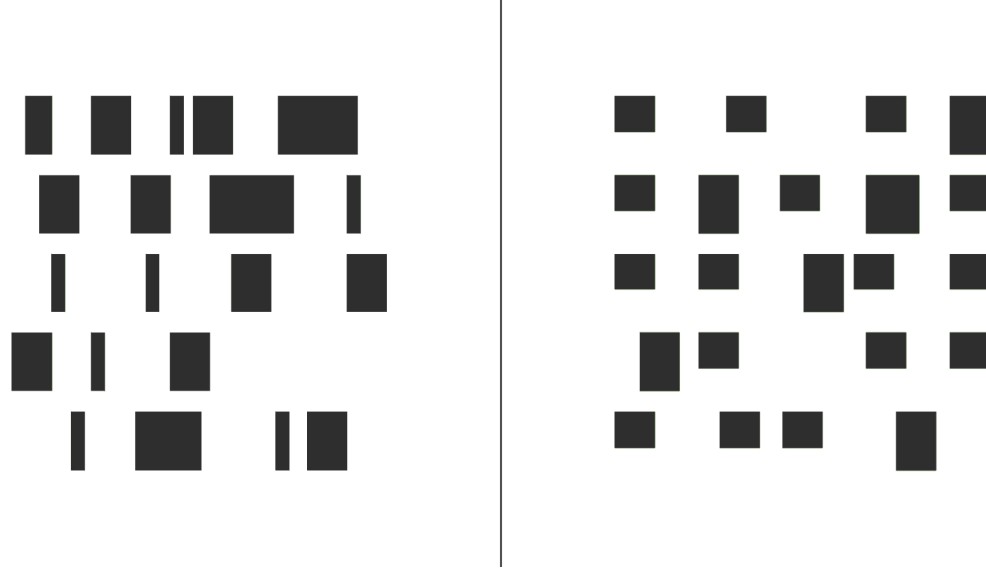

**Figure 7.** Compositions F1 (**left**) and F2 (**right**) presented during the research.

Composition F1 was designed based on three types of composition elements with different widths but the same heights. Composition F2 was also designed based on three

types of compositional elements but with two different widths and two different heights. In the case of both compositions, the horizontal distances between the compositional elements are varied and random; in addition, in the case of composition F2, the vertical distances were also differentiated (two types of vertical distances), marked by the side edges of the compositional elements.

2.2.2. Online Questionnaire

An online questionnaire was created using the QuestionPro.com platform to conduct the survey. The questionnaire had six sections. Access the questionnaire at the following link: https://questionpro.com/t/AQjS0Zlfmd (accessed on 2 May 2023).

**Section 1.** This section included an introduction to the study, including information about the purpose of the study, the time frame for completing the questionnaire, and the complete anonymity of the respondents. An email contact for one of the authors was also provided.

**Sections 2–5.** Participants were asked to rate six visual stimuli on a Likert scale in these four sections. A six-point scale was used due to the corresponding number of composition types. A 1 on the scale indicates the most irregular stimulus, and a 6 indicates the most regular stimulus. The stimuli in each section were presented simultaneously. In each section, the stimuli were presented in a different configuration. In each section, one example of each composition type was presented simultaneously; in other words, no two examples of the same composition type were presented simultaneously. The stimuli were presented in randomized order, as shown in Table 1. Each stimulus was evaluated twice.

**Table 1.** The order in which visual stimuli are displayed in Sections 2–5 of the questionnaire.

| Section/Order No. | 1. | 2. | 3. | 4. | 5. | 6. |
|---|---|---|---|---|---|---|
| Section 2 | A1 | C2 | F1 | B2 | D2 | E1 |
| Section 3 | E2 | A2 | C1 | F2 | B2 | D1 |
| Section 4 | E1 | B2 | C1 | A1 | D2 | F2 |
| Section 5 | C2 | F1 | D1 | A2 | E2 | B1 |

**Section 6.** This section included three questions designed to collect information specific to the survey group, i.e., gender, age, and education.

*2.3. Procedure*

The survey was made available to first-year Architecture and Spatial Management students at the Faculty of Architecture at the Gdańsk University of Technology. The online questionnaire was distributed (as an HTTP link) via the e-learning platform and was available from 16 to 19 March 2021. Participation in the survey was completely anonymous and voluntary.

Respondents were asked to complete an online questionnaire described in the Materials subsection. The questionnaire could be completed on any computer or mobile device with a web browser and Internet access.

*2.4. Statistical Analysis*

For statistical analyses, a one-way ANOVA and a post hoc test were chosen as the primary tools for analyzing averages between different groups to look for statistically significant differences. Post hoc tests of one-way ANOVA show differences in means between specific groups and allow them to be precisely identified.

The GLM procedure was also selected for analysis as a tool for verifying repeated results (respondents evaluated each stimulus twice).

**3. Results**

The questionnaires were completed by 66 respondents, of which 18 responses were incomplete and, thus, were discarded. The study included 48 (N = 48) people aged 19 to 55 years

(M = 22.60; SD = 7.75), of whom 33 were women (68.75%) and 15 were men (32.25%). In total, 41 people (85.42%) had secondary education, and 7 (14.58%) had higher education.

The Cronbach coefficient for a total of 24 composition ratings according to a 6-point Likert scale was .93, which means that the test's reliability is excellent.

### 3.1. Research Question 1

To answer Research Question 1 (is there a relationship between the type of facade composition and its degree of regularity?), a one-way ANOVA was carried out in a one-way variant between groups. The dependent variable (DV) is the degree of regularity, on a scale from 1 (the least regular) to 6 (the most regular), and the independent variable (IV) is the type of composition (described with letters from A to F) to which the presented stimuli belonged. This test is designed to show whether there are statistically significant differences in the average level of visual regularity depending on the type of composition. Table 2 summarizes the results obtained during the above process.

**Table 2.** Summary of ANOVA for the variable "degree of regularity" depending on the type of composition.

|  | SS | df | MS | F |
|---|---|---|---|---|
| Between groups | 2636.87 | 5.00 | 527.37 | 480.56 ** |
| Within groups | 1204.97 | 1.10 | 1.10 |  |
| Total | 3841.84 | 1.10 |  |  |

** $p < 0.001$.

The analysis of variance proved statistically significant: $F_{(5.1)} = 480.56$, $p < 0.001$, $\eta 2 = 0.686$, where the composition's regularity level is a function of its type.

Post hoc tests were performed to determine which composition types differed significantly in their degree of regularity. The assumption of equality of variance was rejected, and Dunnett's T3 test was used as the result of Levene's test for DV was statistically significant ($p < 0.001$). The analysis results show a statistically significant difference ($p = 0.05$) between all types of compositions, except for types B and D.

### 3.2. Research Question 2

To answer Research Question 2 (which types of composition are the most regular and which are the least?), descriptive statistics were carried out, which are presented in Table 3. The DV in these calculations is the degree of regularity (on a scale from 1—the least regular—to 6—the most regular), and the IV is the type of composition (described by letters from A to F) to which the presented stimuli belonged. The results of the analyses clearly show the gradation of the regularity of the composition according to its type. The most regular is type A (M = 5.842; SD = 0.470), and the second is type C (M = 5.163; SD = 0.903). Although the descriptive statistics show differences between the degree of regularity of composition types B (M = 4.054; SD = 1.296) and D (M = 3.696; SD = 1.404), these differences are not statistically significant, as shown via the post hoc tests performed (see the results for Research Question 1). Therefore, it can be assumed that the compositions of type B and D are ex aequo third in the ranking of the degree of regularity. Type E is in fourth place (M = 2.228; SD = 1.072). Type F compositions are considered the least regular (M = 1.402; SD = 0.804).

### 3.3. Research Question 3

To answer Research Question 3 (does the degree of regularity of the facade composition depend only on the type of composition?), a one-way ANOVA was performed between groups. The DV is the degree of regularity (on a scale from 1—the least regular—to 6—the most regular), the IV is the subtype of the composition, i.e., an example of a composition of a given type, described by a combination of letters denoting the general type of the composition (from A to F) and a numerical value indicating the version of the composition of a given type (1 or 2). This test is designed to show whether there are statistically

significant differences in the average level of visual regularity depending on the subtype of composition. A summary of the results obtained during the operation described above is given in Table 4.

**Table 3.** Descriptive statistics for the "degree of regularity" DV against composition types.

|  | Type A | Type B | Type C | Type D | Type E | Type F |
|---|---|---|---|---|---|---|
| Standard Deviation | 0.470 | 1.296 | 0.903 | 1.404 | 1.072 | 0.804 |
| Confidence | 0.068 | 0.187 | 0.117 | 0.203 | 0.155 | 0.116 |
| Standard error | 0.035 | 0.096 | 0.060 | 0.104 | 0.079 | 0.059 |
| Min. value | 5.808 | 3.959 | 5.104 | 3.592 | 2.149 | 1.343 |
| Mean value | 5.842 | 4.054 | 5.163 | 3.696 | 2.228 | 1.402 |
| Max. value | 5.877 | 4.150 | 5.223 | 3.799 | 2.307 | 1.461 |

**Table 4.** Summary of ANOVA for the variable "degree of regularity" by composition subtype.

|  | SS | df | MS | F |
|---|---|---|---|---|
| Between Groups | 2703.35 | 11.00 | 245.76 | 235.72 ** |
| Within Groups | 1138.49 | 1.09 | 1.04 |  |
| Total | 3841.84 | 1.10 |  |  |

** $p < 0.001$.

ANOVA proved statistically significant: $F_{(11.1)} = 235.72$, $p < 0.001$, $\eta2 = 0.704$, demonstrating the relationship between the level of regularity and the subtype of composition (a particular, exemplary composition).

Post hoc tests were carried out to determine which composition examples differed significantly concerning the degree of regularity. The assumption of the equality of variance was rejected as the result of Levene's test for the variable "degree of regularity" was statistically significant ($p < 0.001$), and Dunnett's T3 test was used. There were statistically significant ($p = 0.05$) differences between all pairs of elements except A1:A2, B1:B2, B1:D2, B2:D2, E1:E2, and F1:F2. Descriptive statistics were also performed on the "degree of regularity" variable as a function of composition subtype. Tables 5 and 6 show the results of these statistics.

**Table 5.** Distribution of DV in relation to the stimuli A1, A2, B1, B2, C1, C2.

|  | A1 | A2 | B1 | B2 | C1 | C2 |
|---|---|---|---|---|---|---|
| Standard Deviation | 0.490 | 0.452 | 1.309 | 1.291 | 0.713 | 1.002 |
| Confidence | 0.100 | 0.092 | 0.267 | 0.264 | 0.119 | 0.205 |
| Standard Error | 0.051 | 0.047 | 0.136 | 0.135 | 0.061 | 0.104 |
| Min. value | 5.797 | 5.790 | 3.907 | 3.931 | 5.352 | 4.809 |
| Mean value | 5.848 | 5.837 | 4.043 | 4.065 | 5.413 | 4.913 |
| Max. value | 5.899 | 5.884 | 4.180 | 4.200 | 5.474 | 5.017 |

**Table 6.** Distribution of DV in relation to the stimuli D1, D2, E1, E2, F1, F2.

|  | D1 | D2 | E1 | E2 | F1 | F2 |
|---|---|---|---|---|---|---|
| Standard Deviation | 1.377 | 1.251 | 0.889 | 1.190 | 0.802 | 0.810 |
| Confidence | 0.281 | 0.256 | 0.182 | 0.243 | 0.164 | 0.165 |
| Standard Error | 0.144 | 0.130 | 0.093 | 0.124 | 0.084 | 0.084 |
| Min. value | 3.052 | 4.065 | 1.907 | 2.332 | 1.340 | 1.296 |
| Mean value | 3.196 | 4.196 | 2.000 | 2.457 | 1.424 | 1.380 |
| Max. value | 3.339 | 4.326 | 2.093 | 2.581 | 1.507 | 1.465 |

There are statistically significant differences ($p = 0.05$) between the compositions C1 (M = 5.413; SD = 0.713) and C2 (M = 4.913; SD = 1.002) and between D1 (M = 3.052; SD = 1.377) and D2 (M = 4.065; SD = 1.251). In addition, in the case of the D2 composition (M = 4.196; SD = 1.251), the degree of regularity is greater than the general degree of regularity of type B (M = 4.054; SD = 1.296) and at the same time greater than the degree of regularity of subtypes B1 (M = 4.043; SD = 1.309) and B2 (M = 4.065; SD = 1.291).

### 3.4. Research Question 4

To answer Research Question 4 (has the presentation of stimuli in different configurations influenced the determination of the degree of regularity?), the GLM procedure of repeated measurements was carried out with the assumption of two levels (first and second measurement). The Bonferroni post hoc LSD test was used to evaluate the differences in the analyzed parameters. The statistical analyses showed a statistically significant difference between the results obtained for the first and second samples (i.e., for the first and second stimulus presentations) (MD = 0.071; $p = 0.05$). Two post hoc Bonferroni tests were then performed to detect differences for composition types and subtypes. The results of the above tests showed statistically significant ($p = 0.05$) differences in the first and second measurements between all composition types, except for the B:D pair. The post hoc test on the repeated measurements obtained for the composition subtypes showed statistically significant differences for all pairs except A1:A2, A1:C1, A2:C1, B1:B2, B1:D2, B2:D2, C1:C2, E1:E2, E1:F1, E1:F2, and F1:F2.

For this reason, the between-groups variant of the one-way ANOVA was repeated. In these calculations, the DV is the degree of regularity (on a scale from 1—the least regular—to 6—the most regular) for the first and second measurements. The IV is the type of composition (described by letters from A to F) to which the presented stimuli belonged. The analysis of variance was statistically significant for the first measurement $F(5.551) = 231.16$, $p < 0.001$, and the second measurement $F(5.551) = 248.98$, $p < 0.001$. Levene's test results for all variables were significant ($p < 0.001$), rejecting homogeneity and using Dunnett's T-test. The post hoc test analysis for the first and second measurements showed statistically significant ($p = 0.05$) differences in the degree of regularity for all pairs of composition types except B:D, which is consistent with the results presented earlier, described in the Results—Research Question 1 subsection.

A univariate analysis of variance (ANOVA) was conducted within an intergroup design to examine differences in the DV—the degree of regularity, which was rated on a scale ranging from 1 (minimally regular) to 6 (maximally regular), across two time points, as well as differences based on the IV—composition subtype (a categorical variable represented by combinations of letters A–F and numbers 1–2). The results of the ANOVA revealed a statistically significant effect for the degree of regularity at the first time point, $F(11.551) = 111.26$, $p < 0.001$, and at the second time point, $F(11.551) = 123.93$, $p < 0.001$. Levene's test for equality of variances for the degree of regularity indicated significant violations of the homogeneity of variance assumption at both time points ($p < 0.001$), leading to the rejection of this assumption. Consequently, Dunnett's T3 post hoc test was employed for pairwise comparisons. The analyses comparing the degree of regularity across different composition subtypes, both in the aggregated results as presented in the "Results—Research Question 1" section and separately for the first and second time points, are detailed in Table 7, highlighting the absence of statistically significant differences among the composition subtypes.

Thus, discrepancies were noted in the context of statistically significant differences ($p = 0.05$). The B1:D1 and D1:D2 pairs differed significantly in the degree of regularity in the overall results and the second sample. The C1:C2 pair differed only in the overall results.

**Table 7.** Comparison of the results of the one-way ANOVA in terms of the lack of statistically significant differences in the degree of regularity between the pairs of compositions for the sum of both measurements and separately for the first and second measurements.

| | A1 A2 | A1 C1 | A2 C1 | B1 B2 | B1 C2 | B1 D1 | B1 D2 | B2 C2 | B2 D1 | B2 D2 | C1 C2 | C2 D2 | D1 D2 | D1 E2 | E1 E2 | E1 F1 | E1 F2 |
|---|---|---|---|---|---|---|---|---|---|---|---|---|---|---|---|---|---|
| 1 + 2 | x | | | x | | | x | | | x | | | | | x | | |
| 1 | x | x | x | x | | x | x | x | x | x | x | x | x | x | x | x | |
| 2 | x | | | x | x | | x | x | x | x | x | x | | x | x | x | x |

## 4. Discussion

This study ranked the types of compositions according to their degree of regularity. For this purpose, a questionnaire survey was conducted, and statistical analyses were performed.

### 4.1. Research Question 1

Statistical examinations addressed Research Question 1, which probes the association between facade composition type and its regularity degree. These analyses elucidate the correlation between the composition type and its regularity level. The findings indicate a variance in the regularity degree among all composition types, barring types B and D, between which the regularity does not exhibit a statistically significant difference. The absence of a statistically significant discrepancy in the regularity degree between types B and D could be attributed to the nuanced distinctions between these composition types. These compositions are defined in entirely different ways and have completely different constraints. At the same time, it can be assumed that the specific stimuli prepared for the study are within the tolerances Morgan wrote about [35].

### 4.2. Research Question 2

A series of descriptive analyses were performed to elucidate Research Question 2 concerning the identification of composition types with the highest and lowest levels of regularity. These analyses revealed a distinct hierarchy of regularity among the composition types. Composition type A was identified as exhibiting the highest degree of regularity, attributed to its uniform elements spaced at equal intervals. In contrast, composition type F was the most irregular, disregarding compositional rules. This includes randomness in the sizes and proportions of elements, as well as their placement and spacing. Following type A in terms of regularity is type C, which can be regarded as a derivative of type A compositions. More precisely, type C represents an amalgamation of two exemplary type A compositions, positioning it as the second most regular composition type.

Although descriptive statistics show that type B compositions are more regular than type D, statistical analyses indicate that these differences are not statistically significant. This is interesting because type D is a composite of two type B compositions. Type A compositions, for example, are considered significantly more regular than type C compositions, a combination of two type A compositions. Such results suggest that a person's ability to judge the regularity of stimuli with a moderate intensity of this property is limited. Judgments are unambiguous for stimuli near the extremes of the regularity scale, but judgments become less differentiated for stimuli near the middle of the scale.

The second least regular composition type is type E, demonstrating that theoretical vertical composition lines are much less legible than a group of harmonious elements arranged in a vertical column. These results clearly correlate with Arnheim's observations on the verticality of buildings [3]. Although abstract graphics were presented in the study, the vertical direction is perceptually perceived as superior to the horizontal.

Therefore, the vertical axes of the composition and the relationships between them are most important in determining the composition's degree of regularity (or irregularity). The

more precise the lines of the composition and the smaller the differences in the horizontal distances between the axes, the more regular the composition is perceived. To some extent, this is also confirmed by the experiment conducted by Friedenberg and Bertamini [30], who proved the supremacy of the vertical direction in perceiving the symmetry of abstract elements.

Referring to the results of the statistical analyses conducted to answer Research Questions 1 and 2 and the cited literature [14–19], facades based on type A and C compositions should provide the most pleasant visual experience. However, it should be noted that they are the most repetitive. Therefore, it is worth asking how to design facades based on these compositions while avoiding monotony. Indeed, when designing based on these types of compositions, according to the theory of Birkhoff and Eysenck [10–12], one should maintain the most significant possible degree of complexity.

*4.3. Research Question 3*

To answer Research Question 3 (does the degree of regularity of the facade composition depend only on the type of composition?), statistical analyses were performed. The results of the calculations prove that the degree of regularity of the subtypes (specific exemplary compositions) of the compositions of types A, B, E, and F is similar (there are no statistically significant differences). Therefore, it can be assumed that regardless of the individual characteristics of specific examples of compositions of type A, B, E, or F, the degree of regularity will be consistent with the degree of regularity of the type as a whole. The opposite relationship has been observed in the context of composition types C and D. Thus, a composition of type D may be more regular than a composition of type B due to individual characteristics.

It should be noted that stimulus D2 as a composition consists of two layers (Figure 8), according to the multilayer composition model [38], where individual elements form a D type composition. Still, as groups, they create an A type composition. The statistical analyses of the degree of regularity in composition subtypes also revealed differences between D2, B1, and B2 compositions. Moreover, the descriptive statistics show that the stimulus D2 was classified as more regular than the type of composition B. Therefore, studies based on a more significant number of stimuli from the group of types of compositions B and D would show a difference between these types and, simultaneously, a greater regularity of B types relative to D types.

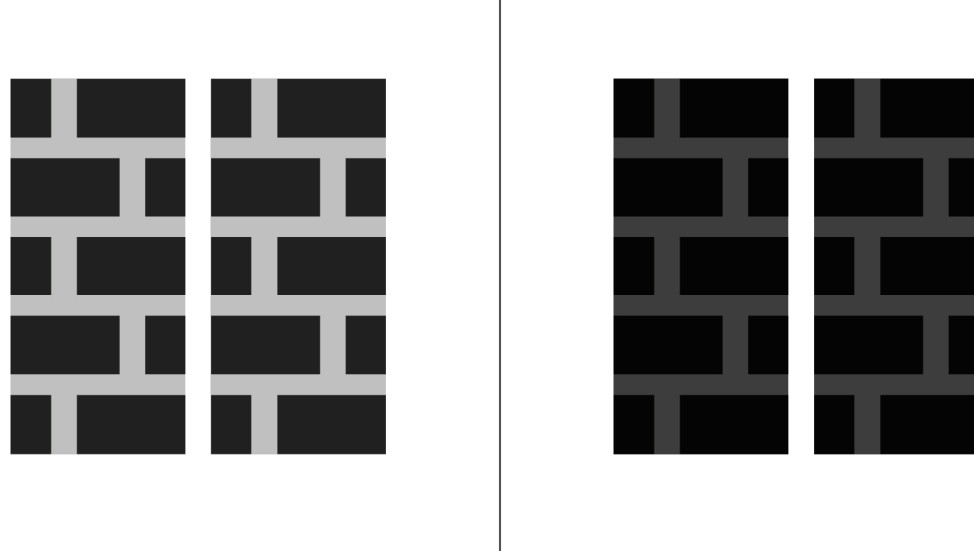

**Figure 8.** Division of the D2 stimulus into compositional layers—type D (**left**) and A (**right**).

Nevertheless, the degree of regularity in this context may depend not only on the type of composition itself but also on the types of individual compositional layers and

individual features. Based on the results presented, this is only true for compositions C and D. These results somewhat agree with previous studies that demonstrate a relationship between regularity and features such as element size or distances between elements [33]. The type of composition, on the other hand, defines how the elements are organized rather than the elements themselves.

*4.4. Research Question 4*

To answer Research Question 4 (has the presentation of stimuli in different configurations influenced the determination of the degree of regularity?), statistical analyses were performed. The results show that the determination of the global degree of regularity of the composition type is independent of other factors. It can be concluded that, for example, type A compositions will always be perceived as more regular than type C compositions.

Nevertheless, stimulus presentation conditions are statistically significant in determining the regularity of specific compositions. Thus, compositions appear regular depending on which other composition they are combined with.

The statistical evaluation of the degree of regularity, segmented by types and composition subtypes across the initial and subsequent measurements, unequivocally demonstrated that the configuration of stimulus presentation exerts a statistically significant influence on the appraisal of stimulus regularity. This phenomenon aligns with findings from prior research, which identified a comparable impact within the context of dot pattern regularity perception, influenced by the regularity degree of the encompassing pattern [19]. Rudolf Arnheim's work [39] offers partial corroboration of this effect, elucidating the principle of similarities and dissimilarities. According to Arnheim, this principle posits that stimuli have the potential to either mitigate or amplify each other's effects, thus affecting the overall perception of regularity.

## 5. Conclusions

To investigate the link between facade composition types [1] and their perceived visual regularity, a study involving 48 anonymous participants was conducted via an online survey. Participants were tasked with assessing visual stimuli—created based on composition definitions by Malewczyk, Taraszkiewicz, and Czyż [1]—using a Likert scale. The gathered data underwent statistical analysis.

Throughout the survey execution and subsequent data analysis, the authors addressed all four posed research questions. However, the findings still need to elucidate the observed phenomena fully. The results presented in Section 3.1 and further discussed in Section 4.1 highlight the ambiguity regarding the regularity of B type and D type compositions. Furthermore, results indicated that the degree of regularity is influenced by the composition type and its unique characteristics and integration with other compositions (see Sections 3.3, 3.4, 4.3 and 4.4). Research is needed to explain the reasons behind these dependencies.

The authors recommend further research to examine the regularity of composition types B and D, how specific compositional techniques affect regularity, and investigate environmental influences on stimulus perception. It may also be beneficial to increase the number of stimuli from the same compositional type to minimize the impact of unique stimulus characteristics on the overall findings for that type. Such expanded research could enhance the understanding of composition perception regarding compositional typology and irregularity levels.

Crucially, this study connected composition type and visual regularity. According to Sections 3.2 and 4.2, compositional types A and C are perceived as the most regular. In contrast, types E and F are considered least regular, with B and D types falling into a moderate regularity category. Given the study's foundation on the universal neurobiological basis of visual regularity perception, the authors suggest that these findings broadly apply to the Polish population. An international study is recommended to extend these

findings universally, although significant variations in results across cultures are yet to be anticipated.

It is vital to note that this study was conducted without considering the multidimensional and interactive nature of architectural perception. For more comprehensive insights, future research should integrate these aspects, answering whether composition defines architecture absolutely or relatively in light of its multifaceted characteristics.

**Author Contributions:** Conceptualization, M.M. and P.C.; methodology, M.M. and P.C.; software, M.M.; validation, A.T. and P.C.; formal analysis, M.M.; investigation, M.M.; resources, M.M.; data curation, M.M.; writing—original draft preparation, M.M.; writing—review and editing, A.T. and P.C.; visualization, M.M.; supervision, A.T.; project administration, M.M., A.T. and P.C. All authors have read and agreed to the published version of the manuscript.

**Funding:** This research received no external funding.

**Data Availability Statement:** The raw data supporting the conclusions of this article will be made available by the authors on request.

**Conflicts of Interest:** The authors declare no conflicts of interest.

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
