# Peer review of "Visual Perception of Regularity and the Composition Pattern Type of the Facade"

_buildings, doi:10.3390/buildings14051389_

Round 1

Reviewer 1 Report

Comments and Suggestions for Authors

I have gone through the manuscript titled "Visual Perception of Regularity and the Composition Pattern Type of the Facade." The authors attempt to analyze the Polish composition pattern type of the façade. In my opinion, the paper faces some shortcomings that need consideration. I would like to suggest the following improvements:

Abstract:

The abstract does not require citations.

Authors need to clarify the research outcomes and provide them.

It is more appropriate to choose different keywords from the title words.

Introduction:

Include more international literature in the review section.

Clarify the meanings of the capital terms within brackets on Lines 45, 47, and 66.

According to the authors (Line 74), what do they mean? Are they referring to the last citation or the authors of this paper? Provide clarification.

Although the authors provided some objectives, it is necessary to organize them based on the identified gaps and literature review.

The first question of the research is answered in the literature review section (Line 140-150). Remove these lines as they are not necessary.

Method:

Introduce the place of study first. How were the authors involved in identifying the 12 types of facade compositions for contemporary Polish residential buildings? Do these 12 types cover the entire spectrum of facade compositions? Provide further reasoning.

Provide a link to the questionnaire (Line 223).

Consider the following points regarding the questionnaire: a. Discuss the adequacy of the student sample. b. Is it a proper sampling method? c. Determine if the respondents are representative of society, and if the study's results are reliable. d. Explain the statistical test used in the method section and the reason for selecting one-way ANOVA.

Results:

Present a brief overview of the respondents in this section.

Mention the total number of completed questionnaires.

Discuss the reliability and validity of the questionnaire.

Discussion:

Focus more on reasoning and case effect analysis rather than being descriptive.

Clarify the authors' points in relation to the theories mentioned in the introduction (e.g., Lines 360-362).

The perception aspect seems to be missing in some parts of the paper, particularly in the research process.

Conclusion:

Change the title to "Conclusion" and remove any additional words.

Focus on the significant outcomes of the research in this section without requiring citations.

Discuss the generalization of the study.

Author Response

Abstract:

- Citation removed from abstract.

- The results of the study are cited and explained.

- Other keywords have been selected

Introduction:

- 7 bibliographic items were added.

- Redrafted the indicated sentences so that it is clear what "O" and "C" are 

- Indicated in line 74 meanings of the authors of this article, supplemented text

- Paragraph 1.3 has been substantially transformed. 

- Removed the indicated passage (lines 140-150).

Method:

- The preliminary description of the stimuli in section 2.2.1 was expanded to clarify the concerns raised. How the authors of the reviewed paper were involved and possibly linked to the article on composition typology was not clarified due to the blinded-review process.

- A link to the questionnaire has been added.

a. Discussed the basis for sample size selection in para. 2.1 b. Discussed in para. 2.1 c. Discussed in para. 2.1 d. Added para. 2.4

Results:

- A description of the respondents with their statistical characteristics can be found at the beginning of section. 3 Results.

- Listed is the total number of completed questionnaires at the beginning of section 3 Results.

- A reliability analysis of the questionnaire was performed, the results are presented at the beginning of section 3 Results.

Discussion:

- The discussion of the results in the discussion, with regard to researcher questions 1-3, has been expanded.

- The description of the study's objectives has been expanded (Section 1.3). The authors intentionally excluded the perception aspect of architecture from the study in order to obtain raw results that can be confronted with the results of broader studies in the future. Such a procedure is ultimately intended to answer the question of which aspects of architecture and how they determine it, and which of them we can analyze in isolation and which not. 

Conclusion:

- The title of Section 5 has been changed.

- Citations have been removed (except for the reference to the item defining the typology of composition - fundamental to the article under consideration). The section has been expanded.

- At the beginning of sec. 5 added a paragraph - discussion of the generalized study.

Reviewer 2 Report

Comments and Suggestions for Authors

Please see the attached review report.

Author Response

- Added para. 2.4 dedicated to the statistical tests that were chosen to analyze the data, collected during the survey. In addition, in para. 3 (Results), describes in detail which analyses, and why, were conducted in the context of the specific research questions.

- This article should be seen as a starting point for further research, which represents further research plans of the authors. The study intentionally neglects the perception aspect and isolates the composition from other architectural elements. Extended para. 1.3 and 5 to clarify this procedure.

- Doubts are raised about composition types B and D - according to the definitions contained in "Composition patterns of contemporary Polish residential building facades."

Reviewer 3 Report

Comments and Suggestions for Authors

This paper's topic, regularity and beauty in building facades, is a well-considered problem in architectural aesthetics. However, this paper does not currently contribute any new knowledge in this area. The proposal did not consider the viewer's position when seeing a building or critical concepts in psychology related to perception (see Space, Time, and Architecture by S. Giedion and The Image of the City by K. Lynch, to name a few). The methodology is clear but may be flawed if the respondents only see the pattern in elevation. Buildings are viewed in perspective with all of the attendant distortions. The study needs to account for this difference. 

Comments on the Quality of English Language

The paper needs moderate editing for English. 

Author Response

Significant changes have been made to the text. The literature review was expanded to include 7 bibliographic items. Section 1.3, devoted to the description of the objectives, was rewritten, including the reasons why the authors decided to isolate the perception aspect from the study. Section 2 was expanded to include a description of the sampling method. The discussion section was rewritten, referring to more bibliographic references, while focusing more on explaining potential reasons for the observed effects. The entire conclusions section was also rewritten.

Round 2

Reviewer 1 Report

Comments and Suggestions for Authors

The revised paper is acceptable; however, the method of research should be briefly indicated in the abstract. Additionally, the method section in each paper should be replicable, including all techniques used throughout the document. I am curious as to why the authors chose to omit Section 2.4: Statistical Analysis. Its inclusion would be beneficial.

Author Response

The abstract was supplemented with a summary of the research method used.
Paragraph 2.4, devoted to a description of the statistical analyses that were used in the treatment of the results, was introduced during the previous round of review comments. 

Reviewer 3 Report

Comments and Suggestions for Authors

The paper improved significantly with the changes, but a few underlying assumptions still need to be tempered. First, "beauty" is a philosophical construct with a historical basis. Statements such as this: 

"In conclusion, it seems reasonable to make the beauty of architecture dependent on (113) visual regularity - according to the theories and sources cited. In the case of buildings, (114) visual regularity is determined by the composition of the elements. " 

are not sound. The authors base this broad conclusion on a particular Western philosophical tradition, from which the assessment of beauty is linked to pleasure through aesthetics (Gr. aesthetikos) or sensation. Their references are primarily nineteenth-century German philosopher-psychologists (or psychophysicists), but their reading of them does not engage with the complexity of those discussions of the relationship between beauty and pleasure. I don't think this obviates the study, but the conclusions must be less universal and broad and more measured and precise. The study shows that visual regularity is statistically significant when measured against non-regularity. Please see also 565-569: this sentence needs to be rewritten for clarity. 

Author Response

The final part of section 1.2 was expanded, indicating the advisability of limiting oneself to Western theories of understanding beauty (the point of reference is the typology developed on the basis of Polish examples of architecture, while Poland culturally belongs to Western countries).

The indicated sentence was reworded, paying more attention to the basis for the conclusions drawn (quoted literature) - lines 108, 109.

Section 5 (Conclusion) has been rewritten, introducing direct references to the results and discussion on which these conclusions are based.

Minor linguistic corrections were made throughout the text to make the conclusions drawn more precise and refer to their sources.

Lines 565-569 (Author Contributions) have been formulated precisely with the guidelines of the journal editors and are not a creative element of the authors.